# Osteogenic Efficacy of Human Trophoblasts-Derived Conditioned Medium on Mesenchymal Stem Cells

**DOI:** 10.3390/ijms231710196

**Published:** 2022-09-05

**Authors:** Yoon-Young Go, Chan-Mi Lee, Sung-Won Chae, Jae-Jun Song

**Affiliations:** 1Department of Otorhinolaryngology—Head and Neck Surgery, Korea University Guro Hospital, Seoul 08308, Korea; 2Institute for Health Care Convergence Center, Korea University Guro Hospital, Seoul 08308, Korea

**Keywords:** human trophoblasts, conditioned medium, osteogenesis, mesenchymal stem cells, mitogen-activated protein kinase cascades, chemokines

## Abstract

Trophoblasts play an important role in the regulation of the development and function of the placenta. Our recent study demonstrated the skin regeneration capacity of trophoblast-derived extracellular vesicles (EV). Here, we aimed to determine the potential of trophoblast-derived conditioned medium (TB-CM) in enhancing the osteogenic differentiation of bone marrow mesenchymal stem cells (MSCs). We found that TB-CM promoted the osteogenic differentiation of MSCs in a dose-dependent manner. Furthermore, it inhibited adipogenesis of MSCs. We also found that the primary trophoblast-derived conditioned medium (PTB-CM) significantly enhanced the proliferation and osteogenic differentiation of human MSCs. Our study demonstrated the regulatory mechanisms underlying the TB-CM-induced osteogenesis in MSCs. An upregulation of genes associated with cytokines/chemokines was observed. The treatment of MSCs with TB-CM stimulated osteogenesis by activating several biological processes, such as mitogen-activated protein kinase (MAPK) and bone morphogenetic protein 2 (BMP2) signaling. This study demonstrated the proliferative and osteogenic efficacies of the trophoblast-derived secretomes, suggesting their potential for use in clinical interventions for bone regeneration and treatment.

## 1. Introduction

Mesenchymal stem cells (MSCs) play a pivotal role in bone metabolism and construction processes via the regulation bone homeostasis and repair of damaged bone. The osteoinductive potential of MSCs in bone healing is of particular interest [1]. In the event of a bone injury, such as a fracture, they recruit immature progenitor cells to the fracture region and stimulate their differentiation into osteoblast cells for bone synthesis. Subsequently, the extracellular matrix (ECM) is secreted on the bone surface by differentiating osteoblasts. This process is known as osteogenesis and responsible for the healing of the fractured bone [2].

Bone morphogenetic protein 2 (BMP2) enhances the osteoconductive activity of MSCs and has been used in both in vitro and in vivo research. It also finds use in the clinical field for the purpose of bone regeneration [3,4]. BMP2 is a well-known cytokine among the osteogenic BMP members of the transforming growth factor-beta (TGF-β) family. It initiates and significantly promotes osteogenic signaling during osteoblastic differentiation [5,6]. Although BMP2 can stimulate the osteogenesis of osteoblasts, its rapid release and tumor-forming potential in the bone defect area are concerning issues that need better understanding before use in high-cost clinical settings [7]. Therefore, in the field of bone regenerative medicine, a novel osteoconductive material needs to be devised and assessed for basic research and clinical feasibility. Fortunately, stem cell biology provides great therapeutic strategies for various MSC-mediated tissue regeneration processes, such as bone healing, skin rejuvenation, and neural repair [8,9,10]. Bioactive materials containing secretomes from stem cells, conditioned medium (CM), and exosomes are considered alternative biomaterials for MSC-mediated bone healing or ex vivo-expanded MSC-based cytotherapy on the bone defect [11,12,13].

The fetus is surrounded by an outer chorionic membrane and inner amniotic membrane. The chorionic membrane plays an important role in the development and protection of the fetus [14]. It consists of an internal trophoblast layer, which plays a critical role in mediating the exchange of nutrients, gases, and hormones between the mother and fetus [15,16]. The biologically active signaling substances, contained in extracellular vesicles (EVs) secreted from the trophoblasts, execute trophoblast functions during pregnancy [17]. Despite the multifunctional properties of trophoblasts, their application in regenerative medicine has not yet been elucidated, compared to other placenta-derived materials, such as amniotic allografts, used for ocular and skin surface reconstruction [18,19].

Our previous studies demonstrated that the extracts from the chorionic membrane have bone-regenerative capacities with significant osteogenic efficacy on osteoblasts [20,21]. This could be attributed to the regenerative activities of trophoblasts present in the chorionic membrane. Numerous bioactive substances secreted from the trophoblasts are packaged into secretomes, which may prove to be a novel therapeutic biomaterial in the field of regenerative medicine. Our recent study found that EVs derived from human trophoblast stem cells (TSCs) show therapeutic capacity in skin regeneration, owing to their proliferation and anti-aging efficacies [22].

In this study, we hypothesized that trophoblast-derived conditioned media (TB-CM) have promoting effects on the osteogenic differentiation of MSCs. The osteogenic efficacy of TB-CM and primary trophoblast-derived conditioned media (PTB-CM) on MSCs were analyzed to test this hypothesis. Treatment with TB-CM promoted osteogenic differentiation of human MSCs (hMSCs), whereas adipogenesis was inhibited. The proliferation and osteogenesis-promoting effects of PTB-CM were also explored. The mechanisms underlying TB-CM-induced MSC osteogenesis were determined using RNA sequencing (RNA-seq), and the roles of cytokine/chemokine-mediated signaling, MAPK cascade, and BMP2 activity were also analyzed. Our findings revealed the role of human TB-CM in promoting the osteogenesis of hMSCs in vitro. They also revealed the role of trophoblast-derived secretomes in the osteogenesis and proliferation of MSCs, making them promising candidates for regenerative interventions in clinical medicine, along with EV-based cytotherapy and cell-free therapy.

## 2. Results

### 2.1. TB-CM Promotes Osteogenesis but Inhibits Adipogenesis in Human MSCs

To investigate the osteogenic effects of human TB-CM on the MSCs, we sequentially performed an assay for ALP activity, qPCR for quantifying the expression of osteogenesis-related genes, and mineralization assay using the Alizarin Red S stain. We also determined the calcium deposition during in vitro osteogenesis of hMSCs in the presence and absence of TB-CM treatment. ALP is a positive marker molecule expressed during the early stages of osteogenic differentiation of MSCs. We found that its activity was significantly elevated in the TB-CM-treated MSCs on day 7 of culture (Figure 1A). Calcium deposition by TB-CM-treated MSCs was thrice that of the OIM control (Figure 2B). During the in vitro osteogenic differentiation of MSCs, the analysis of the expression levels of osteogenesis-specific marker genes, such as *ALP*, *IBSP*, *RUNX2*, osterix (*OSX*), osteocalcin (*OCN*), and osteopontin (*OPN*), showed a dose-dependent increase in the TB-CM- treated groups (Figure 1C). Furthermore, the alizarin red S staining showed a higher mineralization intensity in the TB-CM-treated MSCs than in the cells cultured in OIM only (Figure 1D). These results showed that TB-CM treatment could stimulate the osteogenic differentiation of MSCs.

Further, we analyzed whether TB-CM can stimulate the adipogenic differentiation of MSCs. Our results showed many lipid droplets in differentiated MSCs on day 14, whereas exogenous treatment with TB-CM significantly inhibited the formation of lipid droplets under the same experimental conditions (Figure 2A). In addition, treatment of MSCs with TB-CM during their in vitro adipogenic differentiation led to a dose-dependent decrease in the relative expression of the genes encoding for PPAR-g, adiponectin, and leptin (Figure 2B), which are related to the adipogenic differentiation of MSCs. These results indicate that TB-CM inhibits adipogenesis in MSCs. Collectively, our results show that TB-CM significantly promotes the osteogenic differentiation of human MSCs but inhibits their adipogenic differentiation.

### 2.2. Human PTB-CM Promotes Proliferation and Osteogenic Differentiation of MSCs

To investigate whether the PTB-CM promotes regenerative capacity in the human MSCs, we isolated primary trophoblasts from a post-delivery-discarded human placenta. The cells showing cytokeratin-7 expression were confirmed to be primary trophoblasts (also known as cytotrophoblasts) (Figure 3A). PTB-CM was collected and observed over 15 days of in vitro culture, and the concentration and size of nanoparticles in the CM were analyzed using NTA equipment. The concentration of nanoparticles in the PTB-CM rapidly decreased on day 4, whereas the particle size increased after 4 days of culture (Figure 3B). After 24 h of treatment, the proliferation of MSCs was significantly enhanced by PTB-CM in a dose-dependent manner (Figure 3C and Appendix A). Cytotoxic effects were not observed, even in the cells treated with high concentrations of TB-CM (60%). The MSC proliferation-promoting effects of PTB-CM were compared with those of MSC-CM and TB-CM. The results indicate a significantly higher efficacy of PTB-CM in promoting proliferation, compared to that of the cells cultured with MSC-CM. However, the same does not follow for TB-CM (Figure 3D). Furthermore, we analyzed the effect of PTB-CM in promoting MSC proliferation under serum-free conditions using a CCK8 assay. We found that 20% of PTB-CM supplementation was comparable to cells cultured with serum-containing media. The increase in concentration of PTB-CM, as observed in the early culture days, from 20% to 60%, significantly increased the proliferation rate of MSCs (20% CM, +46%; 40% CM, +68%; 60% CM, +112%), compared with that in the serum-free control cells. However, PTB-CM collected in the early culture days (days 1 to 7) was more efficient for MSC proliferation than the CM obtained in the late culture period (days 8 to 14) (Figure 3E). During in vitro osteogenesis, a significant increase in the ALP activity, calcium deposition, and *RUNX2* gene expression was observed in the MSCs cultured with PTB-CM, compared to that in control OIM cells (Figure 3F). These results showed that PTB-CM affects the proliferation and osteogenic differentiation of MSCs.

### 2.3. Transcriptomic Analysis of the Role of TB-CM during Osteogenesis of MSCs

Next, we analyzed the cellular mechanisms behind the TB-CM-promoted osteogenesis of MSCs. RNA sequencing was performed to determine the changes in the transcriptome levels of the differentiated MSCs upon treatment with TB-CM. The data revealed a significant variation in the mRNA expression levels of differentially expressed genes (DEGs) (<1.5-fold change, *p* < 0.05) on days 3, 7, and 14 of the in vitro osteogenesis of hMSCs, with or without treatment with TB-CM. The significantly upregulated and downregulated DEGs are represented as volcano plots (Figure 4A). The DEG transcriptome profiles were further analyzed using the DAVID and ClueGO bioinformatics tools to investigate the GO enrichment of cellular signaling pathways triggered by the TB-CM treatment during MSC osteogenesis. The 10 GO term enrichments for each GO category, including biological process, molecular function, and cellular components, are represented in order of statistical significance. The key GO terms related to osteogenesis were inflammatory response (GO:0006954), cytokine and chemokine activity (GO:0005125, GO:0008009), BMP2 binding (GO:0036122), signaling receptor binding (GO:0005102), growth factor activity (GO:0008083), and extracellular exosome (GO:0070062) (Figure 4B). KEGG pathway analysis identified several significantly altered pathways at the cellular level in the TB-CM-treated MSCs, including the cytokine–cytokine receptor interaction, tumor necrosis factor (TNF) and interleukin-17 (IL-17) signaling, cell adhesion molecule, mitogen-activated protein kinase (MAPK), and phosphatidylinositol 3-kinase (PI3K)-protein kinase B (Akt) signaling pathways (Figure 4C). ClueGo analysis also supported the results of the GO and pathway analysis, in terms of the cytokine-mediated signaling pathway/immune response and MAPK signaling cascade (Figure 4D).

### 2.4. TB-CM Regulates the MAPK-, BMP2-, and Cytokine-Mediated Cellular Processes during the Osteogenesis of MSCs

MAPK signaling cascade is responsible for the regulation of osteogenesis, a process required for bone development and homeostasis [23,24]. This study supported this role of MAPK signaling, based on the results of the GO and KEGG pathway analyses. To investigate the ability of TB-CM to promote osteogenic differentiation in hMSCs via the MAPK cascade, the activation of this pathway was analyzed using western blotting. The addition of TB-CM to the culture medium of MSCs during osteogenesis significantly induced the phosphorylation of extracellular signal-regulated kinase (ERK) 1/2, p38, and c-JUN N-terminal kinase (JNK) 1/2 in the cells (Figure 5A), indicating that the three MAPK pathways are activated in response to the TB-CM treatment during MSC osteogenesis. Akt signaling was also analyzed in the MSCs cultured in OIM in the presence of TB-CM. The findings were consistent with the results from the previous studies, based on the important role of Akt signaling in bone development and function [25,26]. Upregulation of *BMP2* gene expression and downregulation of the Gremlin-1 (GREM1) encoding gene (BMP2 antagonist) was confirmed by qRT-PCR in the TB-CM-treated MSCs undergoing osteogenesis, indicating that TB-CM might associate the osteoconductive BMP-mediated cellular pathways with the osteogenic differentiation of MSCs (Figure 5B). Further, we selected the upregulated *CXCL* (*CXCL1*, *CXCL3*, *CXCL6*, and *CXCL8*), *CCL* (*CCL7*, *CCL11*, and *CCL13*), and family gene clusters to investigate the expression of cytokines and chemokines in the TB-CM-treated MSCs, confirming their increased expression levels in the TB-CM-treated groups during osteogenesis of MSCs (Figure 5C,D). Significant upregulation of *IL-6* and matrix metalloproteinase (MMP) (*MMP1*, *MMP3*, *MMP8*, and *MMP13*) genes was also observed in TB-CM-treated MSCs (Figure 5D,E). These results showed an increased expression of proteins and genes, associated with the osteogenesis-promoting biological processes, such as the MAPK cascade, Akt, BMP2, and cytokine/chemokine-mediated pathways, as triggered by the TB-CM during osteogenesis in the hMSCs.

## 3. Discussion

Bone remodeling is a dynamic process that occurs throughout the life span of an individual and maintains a balance between bone formation and resorption by regulating osteoblasts and osteoclasts, respectively [27]. With aging, these regulations in the process of bone homeostasis lose their potential significantly, resulting in various bone diseases, including osteopetrosis, osteoporosis, and osteopenia [28]. The abnormal bone remodeling is accelerated in patients with osteoporosis [29].

Bone marrow MSCs not only differentiate into osteoblasts, but also into adipocytes [28]. This indicates that the lineage commitment of MSCs to differentiate into osteoblasts or adipocytes is critical for the regulation of bone remodeling. In certain clinical studies, a significant increase in adipose tissue in the bone marrow was observed in older patients, as well as patients with osteoporosis, as a consequence of dysregulation of lineage commitment at the time of differentiation of MSCs [30]. Our study showed that the CM derived from the human trophoblasts promoted the osteogenesis of MSCs, while inhibiting their adipogenesis. From a therapeutic perspective, TB-CM can prove to be a potential candidate for the treatment of osteoporosis by regulating the balance between osteogenic and adipogenic differentiation of MSCs. Furthermore, TB- and PTB-CM might provide a novel strategy for the prevention of postmenopausal osteoporosis by promoting the osteogenic differentiation of bone marrow MSCs.

Moerman et al. showed that aging affects the differentiation capability of MSCs by downregulating the adipogenesis-related cytokines [31], thus contributing to bone remodeling and upregulating the expression of adipogenesis-related cytokines [32]. Thus contributing to bone remodeling disorders. Our study revealed similar results and showed that cell signal modulators, such as the cytokines, chemokines, growth factors, and transcription factors, regulate the differentiation of MSCs. Using RNA sequencing, we found that the TB-CM treatment of MSCs resulted in an increase in the cytokine production. There was also an upregulation of the *CCL* and *CXCL* chemokine genes during the osteogenesis of bone marrow MSCs. Numerous studies have demonstrated that inflammatory response-related chemokines (C-C and C-X-C-ligands and/or receptors) participate in bone remodeling by regulating the differentiation of osteoblasts and osteoclasts [33,34,35]. The role of cytokines and chemokines in promoting the osteogenesis of MSCs yet remains to be elucidated; however, without a doubt, the inflammatory phase plays a critical role in bone remodeling [36]. In the early stages of bone healing, an acute inflammatory response occurs in the body, which summons the immune cells and bone marrow MSCs to the fracture area for initiating osteogenesis and angiogenesis [36,37]. A recent study proposed that an inflammatory stimulus involving the cytokines could be used to stimulate the bone healing process or induce an adapted bone homeostasis at the injury site [30]. In the present study, an increased expression of the pro-inflammatory genes *CCL* and *CXCL* in the TB-CM-treated MSCs was observed in the early stages of osteogenesis (day 3 to 7), indicating that these might initiate the osteogenesis of MSCs.

Furthermore, we observed an upregulation in IL-17 signaling in the TB-CM-treated MSCs. Several recent studies have indicated the positive effects of IL-17 on bone formation. It has been shown that the IL-17F cytokine induces osteogenesis via the MAPK signaling cascade [31]. Croes et al. reported the function of IL-17 in BMP2-induced ectopic bone formation [38]. IL-17 is recognized as a potent pro-inflammatory cytokine. However, its ability to enhance the immunosuppressive effects of MSCs has also been reported [39]. IL-17, in the presence of MSCs, specifically eliminated ARE/poly(U)-binding/degradation factor 1 (AUF1), which is responsible for degrading the mRNA coding for AU-rich element (ARE)-containing cytokines and inflammatory molecules, consequently enhancing the immunosuppressive ability of MSCs [39]. Pretreatment of MSCs with IL-17 effectively cured liver injury in a concanavalin A-treated hepatitis mouse model, thus demonstrating that IL-17 ultimately enhanced the MSC-mediated therapeutic effects in an inflammatory environment [39].

Among the well-known osteogenesis-related signaling pathways, the significance of the BMP2 and MAPK pathways in the osteogenesis of TB-CM-treated MSCs was analyzed in our study. Following the TB-CM treatment, there was an increase in the levels of phosphorylated JNK and p38 proteins in the osteogenic MSCs, whereas no such finding was observed for ERK. The activation of p38 and JNK is associated with cell differentiation via the MAPK cascade, whereas ERK activation tends to trigger cell proliferation [40,41]. The results show increased *BMP2* gene expression levels and reduced expression levels of *BMP2* antagonists, such as the *GREM2* gene, in the TB-CM-treated MSCs, implicating the relevance of BMP2 signaling in regulating the osteogenesis of TB-CM-treated MSCs. 

We also found that the expression levels of the members of the *MMP* gene family were increased upon treatment of the MSCs with TB-CM in the early stage of osteogenesis. MMPs are proteolytic enzymes present in the ECM and are responsible for the remodeling of the matrix of the bone tissue [42]. Another role of MMPs is deciding the fate of MSCs, such as the proliferation, differentiation, angiogenesis, or migration [43]. During osteogenic differentiation, the downregulation of MMP-1 and MMP-13 in hMSCs suppresses both the ALP activity and mineralization [44,45]. The osteogenic effect of MMP-1 on MSCs is established by the MAPK cascade, including the ERK and JNK pathways [44]. Moreover, MMP-1 and MMP-3 are associated with the migration and proliferation of MSCs and regulators of the differentiation of MSCs during osteogenesis [46,47].

In this study, we found the regenerative capacity of trophoblast-derived secretomes on the osteogenesis of MSCs in the presence of TSCs and PTB. An immortalized trophoblast cell line, isolated from the early first trimester (6–8 weeks), was used as the source of TSCs, due to the ethical challenges in obtaining TSCs from human embryos. Nadi P and colleagues recently showed a similarity between immortalized TSC cell lines and freshly isolated TSCs from blastocytes, in terms of the ability for self-renewal and differentiation, [48] paving the way for future research on the regenerative potential of TSCs. As mentioned above, a limitation of hTSCs is the issue of obtaining the cells and applying them in regenerative medicine. Therefore, we determined the promoting effects of PTB-CM on MSCs, as PTB were isolated from the chorionic tissue in the discarded placenta after the delivery. It is cost-effective and feasible to obtain a large number of human trophoblasts without ethical challenges. The development of proliferative PTBs is necessary for future work on postpartum, placenta-derived PTBs.

## 4. Materials and Methods

### 4.1. Cell Culture Procedure and In Vitro Osteogenic Differentiation

Human trophoblasts (TBs) and bone marrow-derived MSCs were purchased from the American Type Culture Collection (ATCC, Manassas, VA, USA; HTR-8/SVneo, #CRL-3271) and PromoCell (Heidelberg, Germany; #C-12974), respectively. Human TBs were maintained in 5% fetal bovine serum (FBS; Gibco, Grand Island, NY, USA) and penicillin/streptomycin (1000 U/mL; Gibco), containing RPMI-1640 (Gibco) medium. For human MSCs culture, MEM (Gibco) supplemented with 5% FBS (Gibco) and antibiotics (penicillin/streptomycin 1000 U/mL; Gibco). All cells were incubated in a humidified atmosphere at 37 °C and 5% CO_2_.

For the analysis of the in vitro osteogenic differentiation of hMSCs, cells (1 × 10^5^ cells/well) were seeded in 24-well plates with the culture medium. The growth medium was replaced every 2 or 3 days with an osteogenic induction medium (OIM) when the cells reached more than 90% confluency. OIM contained an additional 10 nM dexamethasone (Sigma, St. Louis, MO, USA), 10 mM β-glycerol phosphate (Sigma), and 0.2 mM ascorbic acid (Sigma) in the growth medium.

### 4.2. Preparation of PTB-CM

Human primary trophoblasts (PTBs) were isolated from the human placenta after the approval of the Institutional Review Board (2016GRO0141) at the Korea University Guro Hospital (Seoul, Korea), with informed consent from the patients. The protocol used in a previous study to isolate primary trophoblasts (villous cytotrophoblasts) from the placenta was followed [49]. Full-thickness sections of 2–3 cm were excised from the placenta within 30 min of delivery, and the villous tissue was washed and sliced into small pieces. Blood vessels were removed from the villous tissue using a microscope slide, finely minced using scissors, and three digestion processes were performed with DNase (Roche Diagnostics, Indianapolis, IN, USA) and trypsin (Gibco). Supernatants were collected during villous tissue digestion and centrifuged with FBS. Crude isolated TBs were visible between the layers of FBS and red blood cells in a 50 mL conical tube. After trophoblast collection, density gradient centrifugation was performed for trophoblast fractionation. PTBs were isolated from the opaque band of the density gradient media and either plated for experiments or slowly frozen in liquid N_2_ until use (Appendix A).

Human PTBs were cultured in IMDM supplemented with 5% FBS (Gibco) and 1% penicillin/streptomycin (Gibco). The cells were incubated for 10 days, and the conditioned medium (CM) was collected every day and stored at −80 °C until use.

The conditioned media derived from the human TBs and MSCs were collected when the cells were incubated in the growth medium for 24–48 h, until the cell confluency reached 80%. All conditioned media were filtered with a 0.22 μm syringe filter (Millipore, Billerica, MA, USA) before use.

### 4.3. Alkaline Phosphatase (ALP) Assay

ALP activity was evaluated using a SensoLyte^®^
*p*NPP Alkaline Phosphatase Assay kit (ANASPEC, Fremont, CA, USA), according to the manufacturer’s protocol, after three and/or seven days of in vitro osteogenic differentiation. The color changes, due to ALP activity differences, were measured using a microplate reader at a wavelength of 405 nm.

### 4.4. Calcium Assay and Alizarin Red S Staining

Cells were decalcified using 0.6 N HCl after 14 days of in vitro osteogenic differentiation, and the supernatants were subjected to a calcium assay using a QuantiChrom™ Calcium Assay kit (DICA-500; BioAssay Systems, Hayward, CA, USA), according to the manufacturer’s instructions. The calcium deposition rate was determined using a microplate reader at a wavelength of 612 nm.

For alizarin red S staining, MSCs were fixed in 4% paraformaldehyde for 15 min, and then the staining was performed by treating them with Alizarin Red S solution (Millipore, Darmstadt, Germany) for 20 min. The stained cells were then washed thrice with distilled water, and the color change was observed. Cetylpyridinium chloride (100 mM; Sigma) was added for destaining, and the degree of mineralization was analyzed using a microplate reader at a wavelength of 570 nm.

### 4.5. CCK8 Assay

MSCs were seeded in 96-well plates (density, 1 × 10^4^ cells/well) and added to PTB-CM in a serum-free medium for 24 or 48 h under the specified conditions. The rate of MSC proliferation was determined using a CCK8 kit (Dojindo Laboratories, Kumamoto, Japan), according to the manufacturer’s protocol. The absorbance of the live cells was measured at 450 nm using a microplate spectrometer.

### 4.6. In Vitro Adipogenic Differentiation

MSCs were seeded into 6-well plates (density, 2 × 10^4^ cells/well) and incubated at 37 °C in 5% CO_2_. After incubation for 72 h, the growth medium was removed and added to the adipogenic induction medium (AIM; Lonza, Walkersville, MD, USA), with or without TB-CM. AIM was changed every 2–3 days during the in vitro adipogenesis of hMSCs. The cells were washed with PBS after 14 days of culture and stained with oil red O stain (Sigma). A quantitative reverse transcription-polymerase chain reaction (qRT-PCR) for the adipogenesis-specific peroxisome proliferator-activated receptor-gamma (PPAR-g), adiponectin, and leptin genes was performed. Oil red O staining images of cells from three independent experiments were observed using a cellSens imaging camera (ver.1.18) and software. The area of the cell stain was quantified using ImageJ software.

### 4.7. Nanoparticle Tracking Analysis (NTA)

The size and concentration of the nanoparticles in the PTB-CM were analyzed using a NanoSight™ LM10-HS10 system (NanoSight, Amesbury, UK), which has a combination of a monochromatic laser beam (of wavelength 405 nm) and NanoSight™ tracking software version 3.0. Dilution of samples with PBS (ratio, 1:10) was performed three times in 30 s, and then the average nanoparticle size and concentration were analyzed.

### 4.8. Immunofluorescence Staining

PTBs were seeded in 6-well plates (density, 3 × 10^5^ cells/well) and incubated for 48 h. The cells were fixed in 4% paraformaldehyde for 20 min and permeabilized with 0.4% Triton-X for 5 min. Cells were then soaked in 0.3% bovine serum albumin (BSA; Sigma) and incubated with a primary antibody against cytokeratin-7 (Santa Cruz Biotechnology, Dallas, TX, USA, used at a ratio of 1:200) overnight at 4 °C. The next day, the cells were washed with PBS and incubated with secondary antibodies (Mouse IgG (488), 1:300; Invitrogen, Carlsbad, CA, USA) for 1 h in the dark. After washing, the cells were mounted using Fluoroshield with DAPI (Sigma), and images were captured using a Zeiss LSM 700 confocal microscope.

### 4.9. RNA Extraction and Quantitative Real-Time RT-PCR

RNA was extracted using TRIzol™ reagent (Invitrogen), and then the reverse-transcribed cDNA was obtained using the PrimeSript™ 1st strand cDNA synthesis kit (Takara Bio, Tokyo, Japan), according to the manufacturer’s instructions. Quantitative real-time PCR was carried out using the Power^®^ SYBR Green PCR Master Mix (Life Technologies Co., Ltd., Woolston Warrington, UK), under the following PCR cycle conditions: 95 °C for 1 min, 40 cycles of denaturation each at 95 °C for 15 s, and annealing-extension at 60 °C for 30 s. The 2^(^^−∆∆Ct)^ method was used to analyze the relative mRNA expression levels, which were normalized to *GAPDH* expression levels. Primer sequences used in this study are listed in Appendix A.

### 4.10. Western Blot Analysis

Cells were lysed using a lysis buffer (Invitrogen) with protease and phosphatase inhibitors (Roche), and protein concentrations in the cell lysates were determined using Quick Star Bradford dye reagent (Bio-Rad, Hercules, CA, USA). Equal amounts of proteins were subjected to SDS-PAGE gel electrophoresis, and then the protein-containing gel was transferred to a PVDF membrane (Millipore). The membranes were blocked with TBST (Bio-Rad) containing 5% (*w*/*v*) skim milk for 30 min and then incubated with primary antibodies overnight at 4 °C. On the following day, the membranes were washed with TBST and incubated with horseradish peroxidase-conjugated secondary antibodies for 1 h at room temperature. The protein expression signals were enhanced using a chemiluminescence detection solution (Bio-Rad) and captured using the Fusion Solo imaging system (Vilber Lourmat, Marne-la-Vallee, France). The full-length original gels and blots were included in the supplemental experimental procedure, and the antibodies used are listed in Appendix A.

### 4.11. RNA-Seq

Total RNA was extracted from the MSCs, either TB-CM-treated or control, during in vitro osteogenesis, and the RNA purity was checked using the Quant-iT™ RiboGreen™ assay kit (Invitrogen, #R11490). Illumina libraries were prepared according to the TruSeq^®^ Stranded mRNA Sample Preparation Guide (Illumina platform, part number 15031047 Rev. E) (RNA integrity number, RIN > 7.0). A transcriptome library was constructed using the TruSeq^®^ Stranded mRNA LT sample prep kit (Illumina, San Diego, CA, USA), according to the manufacturer’s protocol. RNA (1 μg) was reverse-transcribed to cDNA and amplified using SuperScript™ II reverse transcriptase (Invitrogen, #18064014), and a final cDNA library was generated. KAPA Library Quantification kits (qPCR Quantification Protocol Guide (#KK4854; Kapa Biosystems, MA, USA) and TapeStation D1000 ScreenTape (#5067-5582; Agilent Technologies, Santa Clara, CA, USA)) for Illumina Sequencing platforms were used to quantify the generated cDNA library, according to the manufacturer’s protocols. Multiple libraries were mixed together at equal molar ratios and then sequenced on an Illumina NovaSeq platform (Illumina).

Sequences were quality-controlled by removing the adapter and low-quality sequences, determined according to the reference genome of *Homo sapiens* (GRCh38) sequences using HISAT2 v2.1.0, before performing transcriptome analysis of the target regions [50]. After alignment with HISAT2, StringTie v2.1.3b was used to assemble the potential transcripts to quantify the relative gene expression levels [51,52]. The Benjamini and Hochberg algorithm was used for the statistical analyses (*p* < 0.05). Gene ontology (GO) annotations and functional enrichment pathways for significant genes were further analyzed using the Database (accessed on 11 August 2021) for Annotation, Visualization, and Integrated Discovery (DAVID) and Kyoto Encyclopedia of Genes and Genomes (KEGG) [53,54]. ClueGO (Cytoscape software version 3.9.0) was used to visualize the enriched biological processes and pathways of the significant genes [55].

### 4.12. Statistical Analysis

Student’s two-tailed *t*-test and one-way analysis of variance (ANOVA) with post hoc Tukey tests were used for statistical analyses. All figures represent statistical significance at the following *p*-values: * *p* < 0.05, ** *p* < 0.01, and *** *p* < 0.001. All data were obtained from triplicate experiments and are presented as mean ± standard deviation.

## 5. Conclusions

In conclusion, our study showed the osteogenic capacity of human TB-CM on MSCs and regulation of osteogenesis-related signaling, including MAPK, BMP2, and chemokine/cytokine activation. We found that treating aged MSCs with PTB-CM promoted cell proliferation and osteogenesis. Therefore, this study suggests that CM derived from the human trophoblasts can be a potential therapeutic agent for bone fractures and age-related osteoporosis.

## Figures and Tables

**Figure 1 ijms-23-10196-f001:**
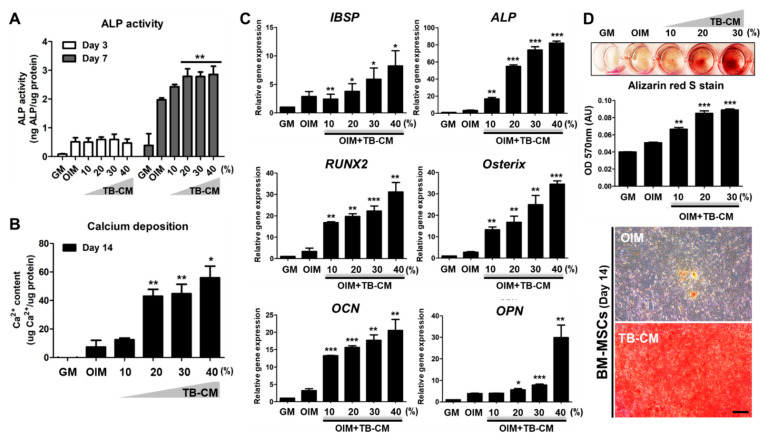
Trophoblast-derived conditioned medium (TB-CM) induces the osteogenic differentiation of mesenchymal stem cells (MSCs). Human MSCs were cultured in varying concentrations of TB-CM in an osteogenic induction medium (OIM) for 14 days. (**A**) In the early stages of osteogenesis (days 3 and 7), the cells were harvested, and the alkaline phosphatase (ALP) activity of each cell lysate was determined and normalized by protein concentration. *p*-values were obtained using the *t*-test, compared to the OIM control. (**B**) Quantification of calcium deposition on the cell surface was performed on day 14. *p*-values were obtained using the *t*-test, compared to the OIM control. (**C**) During the osteogenic differentiation of MSCs, the relative mRNA expression levels of *IBSP*, *ALP*, *RUNX2*, *OSX*, *OCN*, and *OPN* were determined using quantitative reverse-transcription polymerase chain reaction (*IBSP* and *ALP*: day 3, *RUNX2* and *OSX*: day 7, *OCN* and *OPN*: day 14). *p*-values were obtained using the *t*-test and then compared to those of the GM control. (**D**) The matrix mineralization of the MSCs was evaluated by alizarin red S staining, followed by determining the extent of destaining in each culture (arbitrary unit, AU). Representative images of OIM and OIM + 30% TB-CM were taken under a light microscope. Scale bar: 100 μm. Data are presented as mean ± standard deviation (SD) (n = 3); * *p* < 0.05, ** *p* < 0.01, and *** *p* < 0.001.

**Figure 2 ijms-23-10196-f002:**
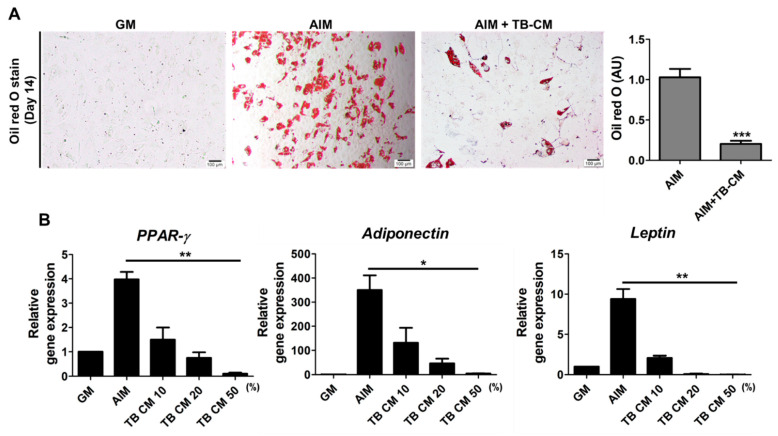
Trophoblast-derived conditioned medium (TB-CM) inhibits adipogenesis of mesenchymal stem cells (MSCs). (**A**) Oil red O staining showing the production of lipid droplets after the adipogenic differentiation of human MSCs upon culturing for 14 days in the adipogenic induction medium (AIM). The inhibitory effect of 50% TB-CM on the adipogenesis of MSCs was determined by the quantification of stained cells, as represented in the graph (AU, arbitary unit; *** *p* < 0.001). (**B**) The relative expression levels of genes encoding for PPAR-g, adiponectin, and leptin in the adipogenic MSCs treated with 10, 20, and 50% TB-CM for 14 days were determined by performing a quantitative reverse transcription-polymerase chain reaction. Data are represented as mean ± SD. *p*-values were obtained using analysis of variance; * *p* < 0.05 and ** *p* < 0.01.

**Figure 3 ijms-23-10196-f003:**
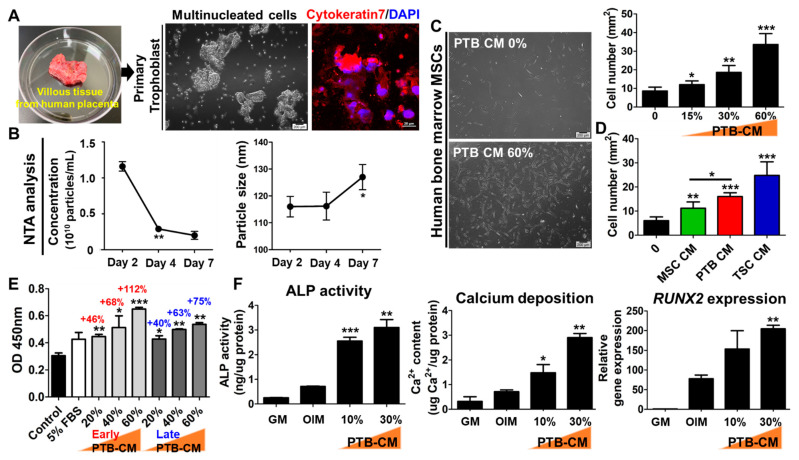
Effect of regenerative ability of primary trophoblast-derived conditioned medium (PTB-CM) on mesenchymal stem cells (MSCs). (**A**) Representative images of human placenta tissue and isolated primary trophoblasts (in vitro cultivation for 72 h). Expression of cytokeratin 7 (red, intracellular marker of human placental villous trophoblasts) was observed as immunofluorescence obtained by nuclear counterstaining with DAPI (blue). (**B**) Conditioned medium obtained from the in vitro-cultured primary trophoblasts was collected every day for 15 days, followed by determining the size and concentration of nanoparticles by performing nanoparticle tracking analysis in the early culture period. *p*-values were obtained using Student’s *t*-test, compared to the day 2 data; * *p* < 0.05 and ** *p* < 0.01. (**C**) Human MSCs (in passage 9) were treated with varying concentrations of PTB-CM for 24 h. Representative images of MSCs treated with 60% PTB-CM and that of the control group were shown, and the cells were quantified to determine the proliferation-promoting effect of PTB-CM. *p*-values were obtained using a *t*-test, compared to that of the untreated cells. (**D**) The proliferation rate of MSCs was compared with the treatment of 30% MSCs-, PTB-, and TSC (trophoblasts stem cells)-derived CM under the same experimental condition. *p*-values were obtained using a *t*-test, compared to the untreated or MSC-CM treated cells. (**E**) MSCs were cultured in a serum-free medium, with an indicated concentration of PTB-CM. PTB-CM was used to divide into early (days 1–7) and late (days 8–14) time collected medium. CCK8 analysis determined the effect of PTB-CM in cell proliferation, indicating an increase in the percentage of cells after PTB-CM supplementation (early, red; late, blue). *p*-values were obtained using a *t*-test, compared to the serum-free control. (**F**) On days 7 and 14 of culturing the osteoblasts in OIM, the effect of PTB-CM on osteogenesis was evaluated by determining the ALP activity, calcium deposition, and *RUNX2* gene expression. *p*-values were obtained using a *t*-test, compared to the OIM group. Data are represented as mean ± SD; * *p* < 0.05, ** *p* < 0.01, and *** *p* < 0.001.

**Figure 4 ijms-23-10196-f004:**
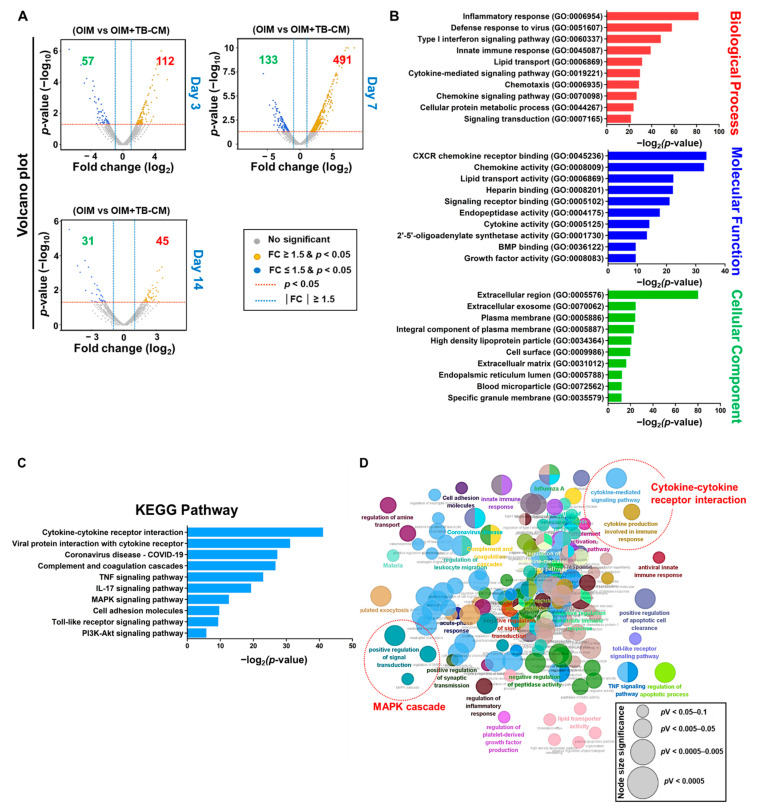
Enrichment analysis of GO terms and KEGG pathways in the TB-CM-treated mesenchymal stem cells (MSCs) during osteogenesis. MSCs were treated with 30% TB-CM under OIM conditions. RNA-seq was performed to identify the differentially expressed genes (DEGs) in the treated MSCs. (**A**) The significantly upregulated and downregulated DEGs are represented in the form of a volcano plot (*p* < 0.05, FC ≥ 1.5). (**B**) DEGs were assigned to three GO categories: biological processes (red bars), molecular functions (blue bars), and cellular components (green bars). The enrichment scores of the GO terms are shown as −log_10_ (*p*-value). (**C**) The ten most significant pathways of DEGs in TB-CM-treated MSCs are presented using a bar chart with enrichment scores shown as {−log_10_ (*p*-value)}. (**D**) Biological processes and pathways of DEGs in the TB-CM-treated MSCs were analyzed using the ClueGO bioinformatics tool. The most significant GO terms and pathways per functional group are shown in bold and in the same color. Node size depicts the level of statistical significance (*p* < 0.05). Consistently analyzed terms, including cytokine-mediated signaling pathways and MAPK cascades, are indicated by red circles. TB-CM, trophoblast-derived conditioned medium; GO, gene ontology; KEGG, Kyoto Encyclopedia of Genes and Genomes; MAPK, mitogen-activated protein kinase.

**Figure 5 ijms-23-10196-f005:**
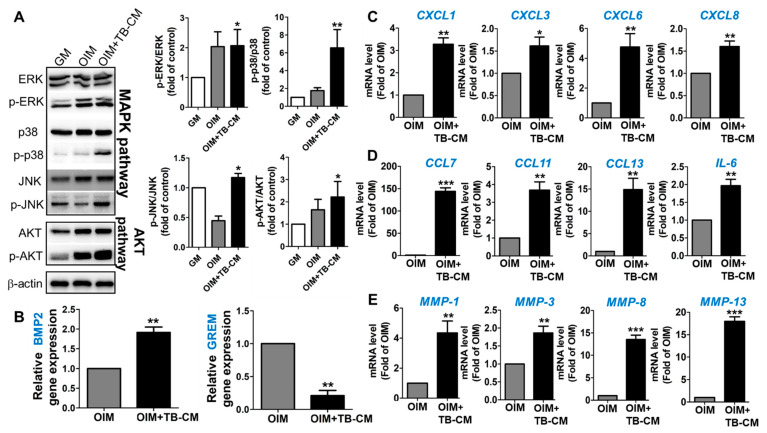
TB-CM induces significant transcriptome alteration in HNDF cells during migration. (**A**) Proteins associated with the MAPK pathway, such as ERK, p-ERK, p38, p-p38, JNK, and p-JNK, were evaluated for their expression levels in the TB-CM-treated mesenchymal stem cells (MSCs) by performing western blot analysis. The expression levels AKT and p-AKT in MSCs treated with TB-CM were also detected using western blot analysis. A quantitative analysis of the western blot was performed, which showed significant expression levels of protein, as presented in the histogram. *p*-values were obtained using a *t*-test, compared to that of the GM group. (**B**) The relative expression levels of BMP2 and GREM1 encoding genes were analyzed both in the TB-CM-treated and control MSCs. A reverse transcription-polymerase chain reaction was performed to confirm the increase in the expression levels of *CXCL* (**C**)-, *CCL* (**D**)-, and *MMP* (**E**)-associated genes in the TB-CM-treated MSCs. Data have been indicated as mean ± SD; * *p* < 0.05, ** *p* < 0.01, and *** *p* < 0.001, compared to the corresponding control.

## Data Availability

All data included in this published article.

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
