# Peer review of "Osteogenic Efficacy of Human Trophoblasts-Derived Conditioned Medium on Mesenchymal Stem Cells"

_ijms, 2022, doi:10.3390/ijms231710196_

Round 1

Reviewer 1 Report

Go and Song et al. demonstrated the potential of trophoblast-derived conditioned medium (TB-CM) in enhancing the osteogenic differentiation of bone marrow mesenchymal stem cells (MSCs). Data showed that TB-CM promoted the osteogenic differentiation of MSCs in a dose-dependent manner and inhibited adipogenesis of MSCs. This study demonstrated that treatment of MSCs with TB-CM stimulated osteogenesis by activating several biological processes and suggest their potential for use in clinical interventions for bone regeneration and treatment.

Some additional comments are given below:

1.     In Figure 1, the author needs to clearly mark the serial number of the panels. 

2.     The font size of Figure 3, 4 and 5 is too small. Authors are advised to increase the font size to make it easier for readers to see.

3.     In Figure 2A, the backgrounds of images are mostly different, and the authors should make adjustments.

Reviewer 2 Report

The manuscript by Go et al. is well-written, sensible, and provides interesting results. I only have two specific comments, one which would improve the section about statistical analyses. Please provide information about how the assumptions for the two-way ANOVA were checked and what post-hoc test (if any) that were used for statistical analyses. Moreover, ® can be omitted from scientific writing.
